# Glucomannan as a Dietary Supplement for Treatment of Breast Cancer in a Mouse Model

**DOI:** 10.3390/vaccines10101746

**Published:** 2022-10-19

**Authors:** Nioosha Ahmadi, Hamid Reza Jahantigh, Hassan Noorbazargan, Mohammad Hossein Yazdi, Mehdi Mahdavi

**Affiliations:** 1Advanced Therapy Medicinal Product (ATMP) Department, Breast Cancer Research Center, Motamed Cancer Institute, Academic Center for Education, Culture and Research (ACECR), Tehran 1999614414, Iran; 2Department of Microbiology and Virology, School of Medicine, Kerman University of Medical Sciences, Kerman 7616913555, Iran; 3Recombinant Vaccine Research Center, Tehran University of Medical Sciences, Tehran 31975, Iran; 4Animal Health and Zoonosis PhD Course, Department of Veterinary Medicine, University of Bari, 70010 Bari, Italy; 5Department of Biotechnology, School of Advanced Technologies in Medicine, Shahid Beheshti University of Medical Sciences, Tehran 1983969411, Iran; 6Biotechnology Research Center, Tehran University of Medical Sciences, Tehran 141556619, Iran; 7Immunotherapy Group, The Institute of Pharmaceutical Sciences (TIPS), Tehran University of Medical Sciences, Tehran 141556619, Iran; 8Department of Immunology, Pasteur Institute of Iran, Tehran 1316943551, Iran

**Keywords:** glucomannan, tumor vaccine, adjuvant, supplement, breast cancer

## Abstract

Konjac glucomannan (KGM) is a water-soluble polysaccharide derived from the Amorphophallus’s tuber and, as herbal medicine has shown, can suppress tumor growth or improve health. However, there has been no investigation into the effects of KGM on breast tumor-bearing mice. Therefore, in two cohort experiments, we assessed the effect of glucomannan at daily doses of 2 and 4 mg for 28 days as a dietary supplement and also glucomannan in combination with tumor lysate vaccine as an adjuvant. Tumor volume was monitored twice weekly. In addition, TNF-α cytokines and granzyme B (Gr–B) release were measured with ELISA kits, and IL-2, IL-4, IL-17, and IFN-γ were used as an index for cytotoxic T lymphocyte activity. Moreover, TGF-β and Foxp3 gene expression were assessed in a real-time PCR test. The results show that glucomannan as a dietary supplement increased the IFN-γ cytokine and Th1 responses to suppress tumor growth. Glucomannan as a dietary supplement at the 4 mg dose increased the IL-4 cytokine response compared to control groups. In addition, cell lysate immunization with 2 or 4 mg of glucomannan suppressed tumor growth. As an adjuvant, glucomannan at both doses showed 41.53% and 52.10% tumor suppression compared with the PBS group. Furthermore, the administration of glucomannan as a dietary supplement or adjuvant reduced regulatory T cell response through decreasing TGF-β and Foxp3 gene expression in the tumor microenvironment. In conclusion, glucomannan as a dietary supplement or adjuvant enhanced the immune responses of tumor-bearing mice and decreased immune response suppression in the tumor milieu, making it a potentially excellent therapeutic agent for lowering breast tumor growth.

## 1. Introduction

Breast cancer ranked first in cancer in 2020 and is the leading cause of cancer fatalities in women worldwide, accounting for one in four cancer cases and one in six fatalities [1]. Although there have been many therapeutic and diagnostic advances, the prevention of breast cancer has not been achieved due to imbalance and alterations in genetic, immunologic, biochemical, molecular, and endogenous factors [1,2]. Breast cancer is generally known as a local disease, but studies show it can metastasize to regional lymph nodes, bone, liver, lungs, and the brain [2,3]. Chemotherapy and radiotherapy are common treatments, but they have deleterious side effects, such as sleep disturbance, vomiting, anxiety, nausea, depression, and fatigue [4,5]. In the advanced stage (stage IV), chemotherapy does not increase median survival [6,7,8].

Recently, herbal medicine has been used to treat patients with or instead of conventional drugs, and reports show benefits and survivability with fewer complications and side effects [9]. Konjac glucomannan (KGM) is a water-soluble, highly viscous polysaccharide derived from the tuber of Amorphophallus of the araceae family [10]. The glucomannan chain has two components: D-mannose and D-glucose, which are attached by β-1,4 linkages in a ratio of 1.6:1. The molecular weight varies from 200,000 to 2,000,000 Daltons [11]. In Asia, some studies have demonstrated the use of glucomannan as a traditional medicine, and it is said to have cured many disorders, including hernia, cough, burns, and breast pain as well as hematological diseases and dermatitis [12,13,14,15,16]. Today, glucomannan is usually associated with nutrition, regulating blood–lipid reduction, and suppressing diabetes and arterial sclerosis [17,18]. Studies have shown the suppression effect of glucomannan on colon, gastric, and hepatic cancers [19,20,21]. Chen et al. discovered that glucomannan prevented the development of gastric cancer cell lines SGC-7901 and AGS in vitro. It had a similar therapeutic effect on a human liver cancer cell line and triple-negative breast cancer (TNBC) cells [22]. Since KGM was determined to be the significant bioactive part of A. konjac, Sawai et al. found that konjac-glucomannan-treated HepG2 hepatic carcinoma cells displayed a substantial reduction in growth that resembled the phenotype of AKe-treated tumor cells [23].

Glucomannan has also been found to act as an antioxidant [13,24,25]. An earlier investigation showed that when separated from C. utilis, it exerted an anti-mutagenic function by reducing ROS levels [26]. Miadoková et al. [26] further concluded that glucomannan from the C. utilis cell wall had an antioxidative effect via iron chelation and scavenging of hydroxyl radicals in mouse leukemia cells. In addition, as a dietary water-absorbing fiber, it has been shown to lower the manufacturing of carcinogens from the gut [27,28,29]. As a result of its hydrolytic capability to turn compounds into carcinogens, β-glucuronidase has been identified as a lysosomal enzyme associated with tumor advancement and metastasis [30].

To improve the effectiveness of existing chemotherapeutic medications and decrease their adverse effects, a specific medicine delivery system is required. Glucomannan displays extremely low toxicity [31]. Additionally, it combines with other bioactive compounds to exert an anti-tumor function [32,33]. Glucomannan is compatible with many other bioactive compounds, and tumor cells quickly take up the conjugated macromolecular complications, thus boosting the healing efficiency of existing anti-cancer compounds.

Moreover, glucomannan has been shown to enhance immune function both in vivo and in vitro [31]. Interleukin-10 (IL-10) is referred to as an immunosuppressive cytokine, which counteracts the apoptotic task of cytotoxic T lymphocytes (CTLs) [34]. It re-energize the immune system to attack the tumor cells by lowering IL-10 levels and promoting the manufacturing of IFN-γ in the tumor [31,35]. Similarly, acidolysis-oxidized konjac glucomannan upregulates the expression of cytokines such as TNFα, interleukin-1β (IL-1β), and IL-6, which collectively strengthen the anti-tumor immune response [36].

In the current study, glucomannan was used as an adjuvant or supplement in a mouse model to unveil its effect on a breast cancer microenvironment. To date, there has been no investigation into the effect of glucomannan on breast cancer in mice.

## 2. Materials and Methods

### 2.1. Cell Culture and Propagation

The 4T1 mouse mammary tumor cell line (National Cell Bank of the Pasteur Institute of Iran, Tehran, Iran) is one of the few breast cancer models that metastasizes efficiently [37]. The cells were grown at the Roswell Park Memorial Institute (RPMI) in a 1640 medium culture containing 10% heat-inactivated fetal bovine serum (FBS) and supplemented with 4 mM L-glutamine, 100 µg/mL streptomycin, 100 U/mL penicillin, 1 mM sodium pyruvate, and 50 µM of a 2-ME solution. The cells were then kept at 37 °C in an incubator with 5% CO_2_. After adequate cell propagation, the cells were harvested using trypsin/ethylenediaminetetraacetic acid (EDTA) (GIBCO, Berlin, Germany), washed three times in phosphate-buffered saline (PBS), and then used for further experiments.

### 2.2. Experimental Mice

Inbred female Balb/C mice were provided by the Pasteur Institute of Iran (Karaj, Iran). The mice were six to eight weeks old and weighed 18–20 g. They were kept at 20–22 °C temperature and in a 12/12 light/dark time cycle in an animal house with enough access to water and food. The experiments were carried out after one week of adaptation to the new environment. All the experiments were carried out according to the animal care protocol of the Pasteur institute of Iran.

### 2.3. Tumor Stock Preparation

To establish a tumor stock, 1 × 106 4T1 cells were injected into the flank of the mice. After the tumor developed and growth reached 20 mm, the samples were used as a stock for tumor induction in the naïve Balb/C mice through surgery.

### 2.4. Tumor Antigen Lysate Preparation

A breast cancer cell line (4T1 cell line) suspension was adjusted to 5 × 10^6^ cells/mL in PBS, and the tumor antigen lysate was prepared by freezing and thawing. Briefly, a cell suspension in PBS containing 1 mM phenylmethylsulfonyl fluoride (PMSF) was disrupted by five freezing–thawing cycles using liquid nitrogen and a water bath at 37 °C, followed by sonication (60 Hz, 0.5 amplitude, 10 cycles). Afterward, the samples were centrifuged at 6000× *g* for 15 min, and the supernatant was harvested. The harvested sample was dialyzed with PBS and passed through a 0.2 µm filter. Then, the protein concentration of the samples was measured by Bradford.

### 2.5. Vaccine Mixture with Glucomannan

Glucomannan powder was purchased from Sigma Company (Sigma-Aldrich, St. Louis, MO, USA). To add it to the tumor lysate vaccine, the first glucomannan powder was dissolved in sterile saline, and then selected doses were added. Herein, based on preliminary and other studies, doses of 2 and 4 mg were mixed by vortexing; thus, for each 100 µg of tumor lysate antigen, 2 or 4 mg of glucomannan was added, and the total volume was adjusted to 400 µL.

### 2.6. Tumor Transplantation to the Experimental Mice

The tumor was separated from tumor-bearing mice and kept in cold PBS that included 100 IU/mL penicillin and 100 µg/mL streptomycin under the laminar hood class II. The tumor was dissected to 3–4 mm and then used for tumor transplantation into the shaved right flank of naïve mice. After 24 h, the mice were anesthetized with ketamine/xylazine injected into the peritoneum. The tumor pieces were then inserted subcutaneously, and the skin was closed with glue stitches after sterilization with alcohol (Figure 1A,B). Two days after transplantation, the mice were used for experiments.

### 2.7. Experimental Groups and Immunization

This study was designed on two platforms. In cohort I, the adjuvant activity of glucomannan in the tumor lysate vaccine model was assessed (Table 1). The first naïve mice were immunized subcutaneously three times at one-week intervals with a different formulation of the vaccine. One week after the final shot, tumor transplantation was performed, and tumor changes were measured twice weekly up to week 4.

In cohort II, glucomannan was used as a dietary supplement, so tumor transplantation was performed on native mice. Then, tumor-bearing mice were divided into four experimental groups and orally administered with doses of 2 or 4 mg of glucomannan daily up to week 4 of the study. Cyclophosphamide was used as the positive control, and PBS as the negative. Tumor volume changes were measured twice weekly.

### 2.8. IFN-γ, TNF-α, IL-2, IL-4, and IL-17 Cytokine Assay

Four weeks after the last immunization of vaccine study (cohort I) and four weeks after oral administration of glucomannan as a dietary supplement (cohort II), blood samples were drawn, and serum samples were isolated and stored at −20 °C for IFN-γ, TNF-α, IL-2, IL-4, and IL-17 cytokines. According to the manufacturer’s manual, the cytokines were measured using commercial ELISA Kits (Mabtech, Nacka Strand, Sweden). The quantity of each was calculated according to its standard curve and reported as pg/mL for each mouse.

### 2.9. CTL Activity

The release of Granzyme B (Gr-B) is considered to be a marker for cytotoxic T lymphocyte (CTL) activity according to the literature [38,39]. Here, 16–18 h after the last immunization in cohort I and the last oral administration in cohort II, serum samples of the mice were prepared and stored at −20 °C for a quantitative measurement of Gr-B using a commercial ELISA Kit (eBioscience, San Diego, CA, USA) according to the company manual. The quantity in an individual mouse was calculated according to the standard curve and reported as pg/mL.

### 2.10. Tumor Growth Measurement

During the study, the tumor volume was measured twice weekly by using a digital caliper and estimated by the formula length/2 × width^2^. For each cohort, tumor measurement was conducted for 4 weeks.

### 2.11. Real-Time PCR Analysis of FOXP-3 and TGF-β Gene Expression in the Breast Tumor Microenvironment

#### 2.11.1. RNA Extraction from Breast Tumors

Tumor samples were removed from the mice, and total RNA was isolated using an RNA extraction kit (CinnaGen, Tehran, Iran). Briefly, 300 mg of tissue was lysed by the addition of 800 µL TRIzol. The upper solution was then transferred to a fresh microtube, after which 200 µL of chloroform was added. The solution was centrifuged at 12,000× *g* for 5 min, and an upper solution containing RNA was transferred to a fresh tube. Then, 180 µL isopropanol was added to the upper solution, vortexed to precipitate the RNA, and then centrifuged at 12,000× *g* for 5 min. The supernatant was removed, and the RNA pellet was attached to the bottom of the microtube. Then, 1000 µL of 70% ethanol was added, vortexed to solubilize the RNA, and centrifuged at 12,000× *g* for 5 min. The supernatant was then removed, and 30 µL deuterium-depleted water (DDW) was added to the RNA pellet. The purified RNA was stored at −70 °C until cDNA synthesis.

#### 2.11.2. Synthesis of cDNA

The cDNA was synthesized by Revert Aid™ First Strand cDNA Synthesis Kit (Fermentas, Vilnius, Lithuania). The reaction mixture contained 1 μg of the extracted RNA, 0.5 μL of a random hexamer primer, 0.5 μL of the oligo dT primer, 1 μL of RNase enzyme inhibitor (20 units/μL), 1 μL of reverse transcriptase enzyme, 5 μL of the 5× reaction buffer, 2 μL of the deoxynucleotide triphosphate mixture (10 mM), and double-distilled water (up to a final volume of 20 μL). The temperature program was set as follows: 25 °C (5 min) → 42 °C (60 min) → 70 °C (5 min) to complete the synthesis.

#### 2.11.3. Real-Time PCR Analysis

The primer sequences for TGF-β, Foxp3, and β-actin are given in Table 2. A Rotor-Gene 6000 (Corbett Life Science, Mortlake, Australia) was used to perform a real-time PCR test containing 20 μL of Master Mix, including 5 μL of diluted cDNA, 10 pmol of forward and reverse primers, and 12.5 μL of SYBR Green-containing Master Mix (Ampriqon, Denmark), in a temperature program as follows: 95 °C (10 min) → 95 °C for 20 s → 60 °C (1 min). If the test is 100% efficient, relative gene expression can be calculated according to the ΔΔCt method explained in the literature [40,41,42]. In our study, the PCR efficiency was 100%. Each sample was normalized to β-actin, an endogenous control, and had a fold change. Every gene was determined through the Ct method.

### 2.12. Statistical Analysis

All the data in this study are presented as means ± standard deviation (SD), and Graph Pad Prism v 6.01 software was used for statistical analysis. Parametric tests were used to analyze data after passing the normality Shapiro–Wilk test. One-way ANOVA was used to calculate differences in the experimental groups. Results from Gr-B release and Foxp3 and TGF-β gene expression were analyzed by a Mann–Whitney U test. In all the analyses, *p <* 0.05 was considered a significant difference.

## 3. Results

### 3.1. TH1 Cytokine Response

Cohort I immunization with cell lysate showed a significant increase in IFN-γ cytokine response compared to the PBS control group (*p* = 0.0039). In addition, admixing cell lysate vaccine with glucomannan at 2 and 4 mg doses significantly decreased IFN-γ cytokine release compared to the cell lysate vaccine group (*p* < 0.0039 and *p* < 0.0063, respectively) (Figure 2A). In cohort II, the tumor-bearing mice fed with glucomannan in 2 and 4 mg doses showed a significant IFN-γ cytokine increase compared to the cyclophosphamide group (*p* < 0.005), while those fed 4 mg glucomannan showed a significant increase compared to the PBS control group (*p* = 0.0038). Cyclophosphamide treatment showed significant inhibition of IFN-γ cytokine release compared to the PBS group (*p* = 0.0006) (Figure 2B). In cohort I, TNF-α cytokine release results indicated that immunization with cell lysate, cell lysate + GM2 mg, and cell lysate + GM4 mg showed an increase in IFN-γ cytokine compared to the PBS group, but it was not statistically significant (*p* > 0.2136) (Figure 2C). In addition, results from TNF-α cytokine release in cohort II indicated that mice fed the 4 mg dose showed a significant increase compared to both the cyclophosphamide and PBS groups (*p* = 0.0055 and *p* = 0.0107, respectively) (Figure 2D). In cohort I, immunization by cell lysate and cell lysate + GM2 mg showed a significant increase in IL-2 cytokine response compared to the PBS control group (*p* = 0.0031 and *p* = 0.0052, respectively) (Figure 2E). In addition, in cohort II, results for the IL-2 cytokine showed that the mice fed the 2 mg dose of glucomannan had a significant increase compared to the cyclophosphamide and PBS groups (*p* = 0.0156 and *p* = 0.0379, respectively). In comparison, the mice fed the 4 mg glucomannan dose showed a slight increase compared to the PBS and cyclophosphamide groups (*p* = 0.9999) (Figure 2F).

### 3.2. TH2, TH17, and CTL Cytokine Response

In cohort I, results for IL-4 cytokine in the experimental groups did not show any significant difference among all the experimental groups (Figure 3A). In cohort II, results for IL-4 cytokine indicated that glucomannan at the 2 mg dose showed a significant increase compared to the cyclophosphamide group (*p* = 0.0367). In comparison, glucomannan at 4 mg showed a significant increase compared to the cyclophosphamide and PBS control groups (*p* < 0.0001 and *p* = 0.0006, respectively). The 4 mg dose of glucomannan showed a significant increase in IL-4 cytokine release over the group given with 2 mg (*p* = 0.0038) (Figure 3B). In cohort I, results from the IL-17 cytokine indicated that immunization by cell lysate + GM2 mg and cell lysate + GM4 mg showed a tiny increase compared to the cell lysate and PBS group. However, the difference was statistically not significant (*p* > 0.1533) (Figure 3C). In cohort II, results for IL-17 cytokine indicated that the 2 mg dose did not show a positive effect compared to the cyclophosphamide or PBS group (*p* = 0.5171 and *p* = 0.9999, respectively). In comparison, the 4 mg dose had a positive effect compared to the cyclophosphamide and PBS control groups (*p* = 0.0037 and *p* = 0.0576, respectively) (Figure 3D). In cohort I, the results for CTL activity based on Gr-B release showed that glucomannan at the 2 mg dose as an adjuvant improved the CTL response and showed a significant increase in comparison to cell lysate + GM4 mg, lysate vaccine, and PBS groups (*p* = 0.0145, *p* = 0.0001, and *p* = 0.0148, respectively) (Figure 3E). In cohort II, the results for Gr-B release showed that mice fed the 2 mg dose showed a 58.71, 11.61, and 78.35% increase compared to treatment with the 4 mg glucomannan, PBS, and cyclophosphamide groups (*p* = 0.1095, *p* = 0.6555, and *p* = 0.0511, respectively). In contrast, immunotherapy with glucomannan at 4 mg did not show a dramatic effect compared to the PBS and Cyclophosphamide groups (*p* = 0.4630 and *p* = 0.3823, respectively) (Figure 3F).

### 3.3. Tumor Growth Change

In cohort I, the results for tumor changes on day 28 after transplantation showed that vaccination with cell lysate, cell lysate + GM2 mg, and cell lysate + GM4 mg suppressed tumor growth by 12.27, 41.53, and 52.10% in comparison to PBS-treated mice (*p* = 0.8410, *p* = 0.0418, and *p* = 0.1309, respectively). Immunization with cell lysate + GM2 mg and cell lysate + GM4 mg suppressed tumor growth by about 35.43 and 26.01%, respectively, compared to the cell lysate vaccinated group (*p* = 0.3708 and *p* = 6052, respectively). Immunization with cell lysate + GM2 mg suppressed about 7.47% of tumor growth in comparison to the cell lysate + GM4 mg group (*p* = 0.9864) (Figure 4A). In cohort II, results from tumor volume changed 4 weeks after oral administration with glucomannan at 2 and 4 mg doses and cyclophosphamide showed 28.00, 11.62, and 41.25% tumor suppression in comparison to the PBS-treated group (*p* = 0.4168, *p* = 0.8742, and *p* = 0.1156, respectively). In addition, administration with cyclophosphamide showed 10.34 and 26.54% tumor suppression compared to glucomannan at 2 and 4 mg doses (*p* = 0.9557 and *p* = 0.5228, respectively). Furthermore, administration with 2 mg glucomannan showed 14.67% tumor suppression compared with the 4 mg group (*p* = 0.8681) (Figure 4B).

### 3.4. Foxp3 and TGF-β Genes Expression in Cohort I

In cohort I, the results for the expression of the genes involved in immune response suppression in the tumor microenvironment showed that immunization with the vaccine with glucomannan at the 2 mg dose suppressed Foxp3 gene expression compared to the lysate vaccine and PBS group (*p* = 0.1508 and *p* = 0.0079, respectively). Immunization with cell lysate + GM4 mg showed a significant suppression of Foxp3 gene expression compared with cell lysate + GM2 mg and PBS groups (*p* = 0.0159 and *p* = 0.0079, respectively). In addition, vaccination with cell lysate significantly suppressed Foxp3 gene expression compared with the PBS group (*p* = 0.0079) (Figure 5A). Results from TGF-β gene expression in the tumor microenvironment showed that immunization with cell lysate + GM2 mg, cell lysate + GM4 mg, and lysate vaccine significantly decreased in comparison to the PBS group (*p* = 0.0079). Immunization with cell lysate + GM2 mg and cell lysate + GM4 mg reduced TGF-β gene expression compared with the lysate vaccine group (*p* = 0.0079). Immunization with cell lysate + GM2 mg did not show any significant difference compared to the cell lysate + GM4 mg group in TGF-β gene expression (*p* = 0.2222) (Figure 5B).

### 3.5. Foxp3 and TGF-β Gene Expression in Cohort II

Results for the expression of the Foxp3 gene in the tumor microenvironment showed that after 4 weeks of oral administration with Cyclophosphamide, there was a significant decrease in Foxp3 gene expression compared to glucomannan at 2 and 4 mg doses and the PBS group (*p* < 0.05, *p* < 0.05, and *p* = 0.0079, respectively). In addition, administration with glucomannan at 2 and 4 mg showed a significant decrease compared to the PBS group (*p* = 0.0079). In contrast, no significant difference was observed between the two glucomannan doses (*p* = 0.9444) (Figure 6A). Results for TGF-β gene expression in the tumor microenvironment showed that after 4 weeks of oral administration with cyclophosphamide, there was a significant decrease compared to 2 and 4 mg glucomannan and the PBS group (*p* = 0.0079). Administration with glucomannan at 2 and 4 mg doses showed a significant decrease compared to the PBS group (*p* = 0.0079 and *p* = 0.0159, respectively) (Figure 6B).

## 4. Discussion

While significant advances in cancer therapy have been achieved in the last decade, breast cancer immunotherapy has encountered several barriers, such as the toxicity effects of chemotherapy drugs on normal tissue and drug resistance [43]. Recently, herbal medicine, because of fewer complications and side effects compared to conventional therapy, has been more encouraging [44,45].

Because of the anti-inflammatory and antitumor effect of glucomannan, we used this compound for the first time to treat breast cancer-bearing mice as an adjuvant in the tumor lysate cancer vaccine model and as a dietary supplement [19,46]. Furthermore, using an autologous tumor lysate as a vaccine antigen was considered reliable against tumor development because it might consist of all the essential epitopes that could boost CD4+ assistant T cells and CD8+ T cells [47,48]. It has shown promising results in in vivo and in vitro examination against pancreatic cancer, melanoma, and breast cancer [49,50,51]. Nonetheless, there is one essential worry concerning the immune inductivity of tumor lysate vaccines: if the expression level of a tumor-associated antigen on the tumor cells is minimal, the lysate originating from such a tumor tissue might not become a reliable vaccine antigen due to its weak immunogenicity. Therefore, tumor lysate alone might not create effective immune responses, so a different approach to enhancing vaccine immunogenicity is needed. For instance, our previous study showed that the cell lysate vaccine combined with propranolol could be effective against breast cancer [52]. In addition, studies showed the suppression effect of glucomannan on tumor cells of different cancers, such as colon carcinoma, gastric cancer, and hepatoma; however, to our knowledge, there has been no investigation into the effect of glucomannan with a cell lysate vaccine in tumor-bearing mice [19,20,21].

Our study showed that glucomannan as an adjuvant in the vaccine formulation did not positively affect IFN-γ and IL-2. The treatment of tumor-bearing mice with glucomannan as a dietary supplement at the 4 mg dose increased the IFN-γ cytokine response, and at the 2 mg dose, it increased the IL-2 cytokine response. A previous study by Chen et al. on glucomannan nanoparticles showed the potency of this adjuvant in inducing IL-2 and IFN-γ and enhancing cellular immune responses in an ovalbumin immunogen model [53]. In addition, a study by Suzuki et al. showed that oral administration of glucomannan reduced plasma IgE but not IgG2a, thus confirming Th1 polarization and preventing the development of dermatitis in mice [54].

In addition, the study of Onitake et al. showed the potency of glucomannan in inducing IFN-γ and Th1 responses in a mouse model of oxazolone-induced colitis [55]. However, glucomannan as an adjuvant suppressed the IFN-γ cytokine response and did not show a positive effect on the IL-2 cytokine response. This controversy in the vaccine response may have been due to the nature of the immunogen, which was different in our experiment. According to the present study findings, glucomannan as a dietary supplement can shift the immune response toward the Th1 pattern, while an adjuvant was not successful in the tumor lysate vaccine model; however, this finding may relate to the nature of the vaccine.

As an adjuvant, glucomannan did not change the IL-4 response significantly, while oral administration of glucomannan as a dietary supplement at the 4 mg dose increased the IL-4 cytokine response compared to that in the control groups. On the other hand, previous studies showed the potency of oral administration in suppressing IL-4 gene expression in vaccinated mice, which was not in agreement with our findings [56]. Cytokine IL-4 as a Th2 characteristic is accompanied by a poor prognosis for breast cancer. However, an increase in the IL-4 cytokine level was observed in the oral administration of breast tumor-bearing mice, and this may have triggered antitumor immune responses by stimulating other immunologic mechanisms through its pleiotropic effects on immune cells [52].

Results of the TNF-α and IL-17 cytokines assessments showed that glucomannan as an adjuvant in the vaccine formulation does not positively affect these cytokines. At the same time, oral administration of glucomannan as a dietary supplement at the 4 mg dose increased TNF-α and IL-17 cytokine responses in tumor-bearing mice compared to the control groups. A study by Onishi et al. on the dietary effect of glucomannan on a model of atopic dermatitis in mice showed a suppression effect on the TNF-α cytokine response [57]. In another study, Rupa et al. showed that glucomannan in an ovalbumin-induced egg allergy in mice reduced IL-17 cytokine levels, but higher IFN-γ and Th1 responses were observed; therefore, it seems to be a proper dietary intervention for allergies [54]. A study by Gurusmatika et al. showed that glucomannan could increase macrophage activity and TNF-α production. Overall, the results from these studies do not agree with our findings in the tumor immunotherapy model, which may relate to the nature of the immunogen they used for their studies. It is clear that IL-17 and TNF-α have an antitumor effect and are essential for suppressing tumor growth [52,58].

In the next step, the CTL activity of experimental mice was evaluated through the assessment of the release of Gr-B. As an adjuvant, glucomannan at the 2 mg dose significantly affected the CTL response. In contrast, as an oral dietary supplement, glucomannan at the 4 mg dose showed an 11.61% increase compared to the PBS group and a 78.35% increase compared to the cyclophosphamide group. The critical role of CTL in killing tumor cells and suppressing their growth was highly evident, and several studies have shown the role of CTL in suppressing and clearing tumor cells [32,33,34]. In the next step, changes in tumor volume were monitored: glucomannan as an adjuvant at 2 or 4 mg with a cell lysate suppressed tumor growth by 35.43 and 26.01%, respectively, compared to the cell lysate vaccine group. In cohort II, oral administration with glucomannan at 2 and 4 mg showed 28.00 and 11.62% tumor suppression compared with the PBS group. Considering the immunologic parameters in each cohort and tumor growth, it seemed glucomannan, as a vaccine adjuvant or as a supplement in oral administration has an antitumor effect, but it was not statistically significant. Several studies have shown the antitumor effect of glucomannan in in vivo and in vitro tumor models and have supported our findings [20,23,28,59]. At the end of the study, the regulatory T cell response was analyzed through TGF-β and Foxp3 gene expression in the tumor microenvironment, which was reduced in both cohorts as a result of the 2 and 4 mg doses of glucomannan. These findings showed the potency of glucomannan in modulating the suppressor environment to an immunogenic environment, thereby improving immune responses against tumors. However, to clarify the effect of glucomannan on TGF-β and Foxp3 targets proteins, more analysis including histopathology and immunohistochemistry is needed.

## 5. Conclusions

Results showed that glucomannan enhanced the immune responses of tumor-bearing mice and decreased suppression activity in the tumor environment, thereby suppressing breast tumor growth. However, it is important to remember that glucomannan has also been recommended for its anti-inflammatory effect [57,60]. Therefore, our investigation and previous reports showed that the immune regulative function of glucomannan is bidirectional and that it depends on the nature of the immunogen, the type of cancer, and the route of delivery.

## Figures and Tables

**Figure 1 vaccines-10-01746-f001:**
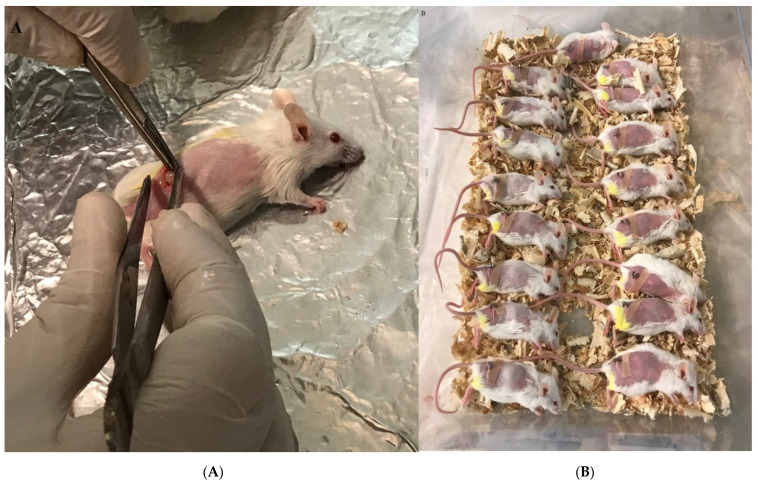
Breast tumor transplant in the experimental mice. (**A**) Tumor transplant into the anesthetized mice in the right flank and subcutaneous insertion. (**B**) Experimental mice after tumor transplant in a recovery situation.

**Figure 2 vaccines-10-01746-f002:**
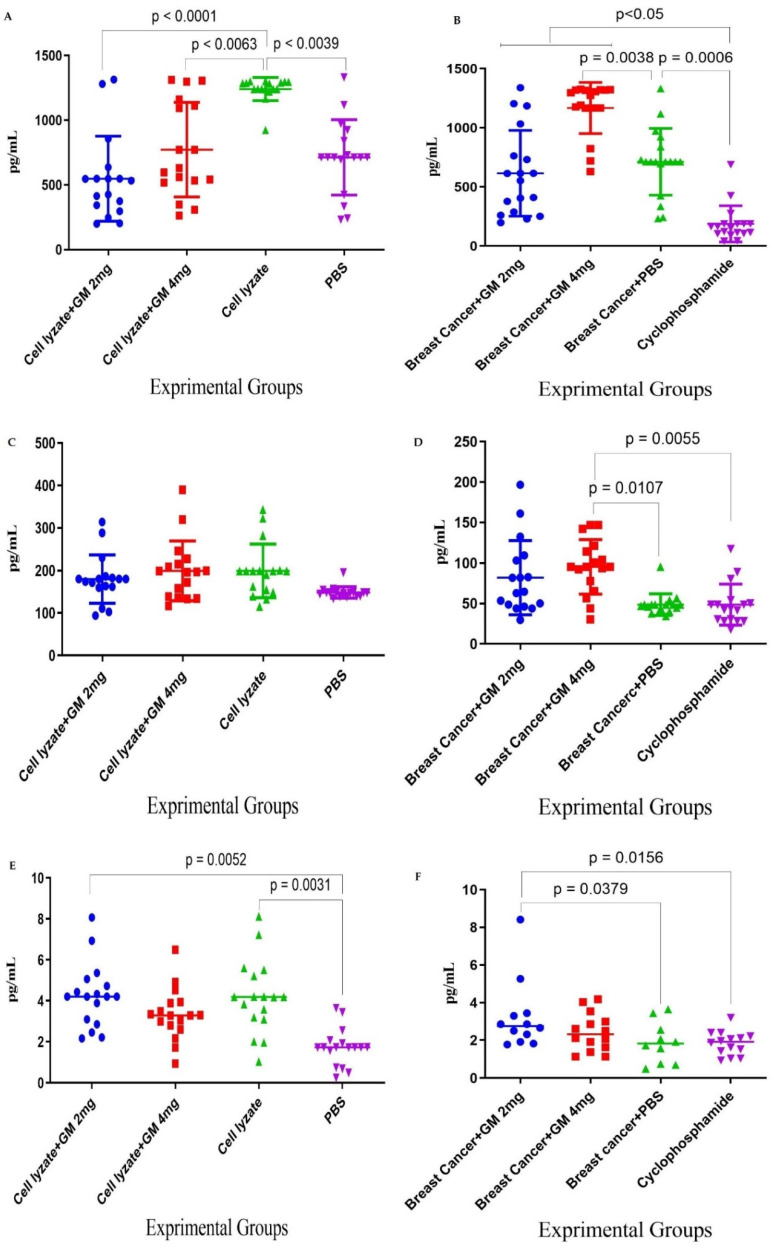
TH1 cytokine response: (**A**) In cohort I, the IFN-γ cytokine response and admixing cell lysate vaccine with glucomannan at 2 and 4 mg doses showed a significant decrease in IFN-γ cytokine release compared with the cell lysate vaccine group (*p* < 0.0039 and *p* < 0.0063, respectively). (**B**) In cohort II, the IFN-γ cytokine response in mice fed the 2 and 4 mg doses of glucomannan showed a significant increase compared to the cyclophosphamide group (*p* < 0.005), while those fed the 4 mg dose showed a significant increase compared to the PBS control group (*p* = 0.0038). (**C**) In cohort I, TNF-α response in immunization with cell lysate, cell lysate + GM2 mg, and cell lysate + GM4 mg showed a slight increase compared to the PBS group (*p* > 0.2136). The glucomannan in the tumor lysate vaccine at 2 and 4 mg did not show a positive effect compared with the tumor lysate vaccine (*p* > 0.8769). (**D**) In cohort II, the TNF-α response with glucomannan at 4 mg showed a significant increase compared to the cyclophosphamide and PBS groups (*p* = 0.0055 and *p* = 0.0107, respectively). (**E**) In cohort I, cytokine IL-2 response in immunization by cell lysate and cell lysate + GM2 mg showed a significant increase compared to the PBS group (*p* = 0.0031 and *p* = 0.0052, respectively). Immunization with cell lysate vaccine with the 2 and 4 mg doses did not show a significant effect compared to the cell lysate vaccine (*p* = 0.9999). (**F**) In cohort II, cytokine IL-2 response in the tumor-bearing mice with glucomannan at 2 mg showed a significant increase compared to the cyclophosphamide and PBS groups (*p* = 0.0156 and *p* = 0.0379, respectively).

**Figure 3 vaccines-10-01746-f003:**
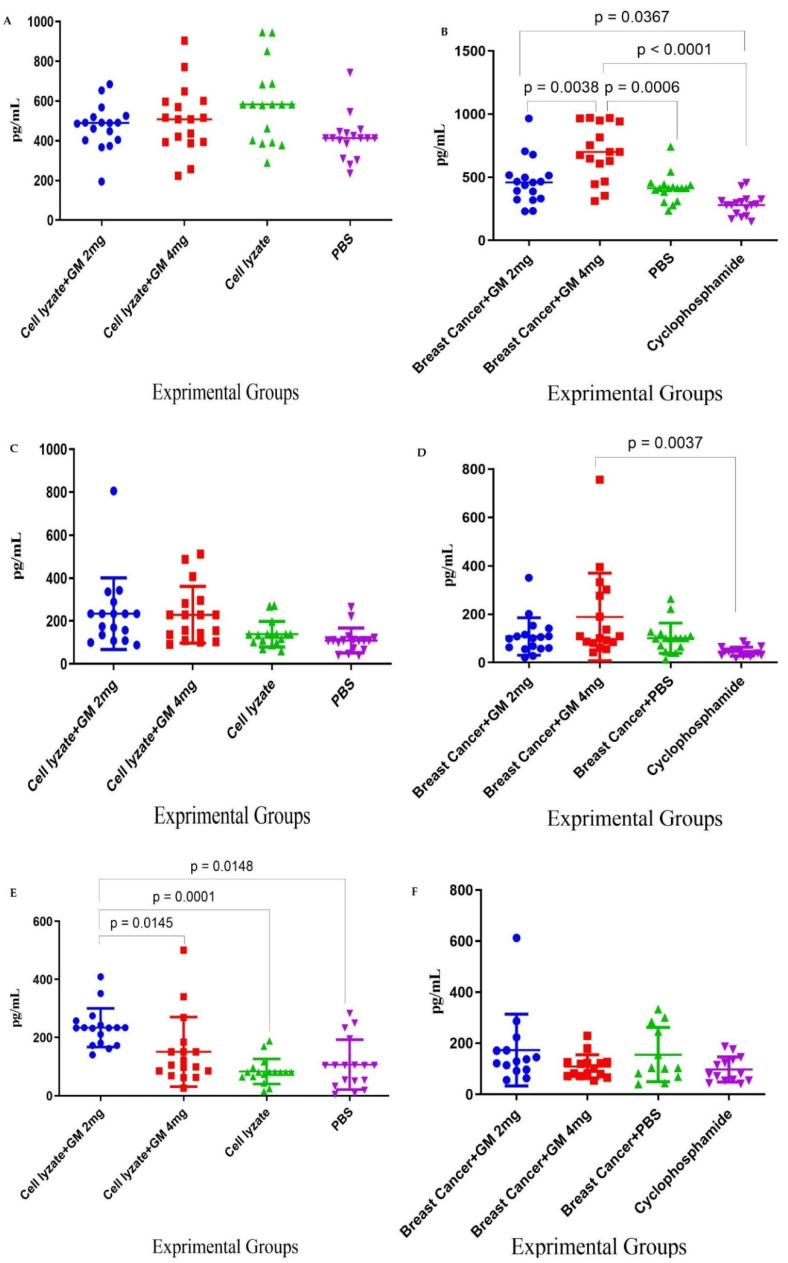
TH2, TH17, and CTL cytokine response: (**A**) In cohort I, IL-4 response in immunization with cell lysate vaccine with glucomannan at 2 and 4 mg doses does not show a significant effect compared to mere lysate vaccine (*p* > 0.1067). (**B**) In cohort II, IL-4 response with glucomannan at the 2 mg dose showed a significant increase compared to the cyclophosphamide group (*p* = 0.0367). In comparison, glucomannan at 4 mg showed a significant increase compared to cyclophosphamide and PBS control groups (*p* < 0.0001 and *p* = 0.0006, respectively). Treatment with 4 mg of glucomannan showed a significant increase in IL-4 cytokine release compared to the group that was treated with 2 mg (*p* = 0.0038). (**C**) In cohort I, IL-17 cytokine response in immunization by cell lysate + GM2 mg and cell lysate + GM4 mg showed a slight increase compared to the cell lysate and PBS groups (*p* > 0.1533). (**D**) In cohort II, IL-17 cytokine response with glucomannan at the 2 mg dose did not show a positive effect compared to the cyclophosphamide and PBS groups (*p* = 0.5171 and *p* = 0.9999, respectively). In comparison, the 4 mg group showed a positive effect compared to the cyclophosphamide and PBS control groups (*p* = 0.0037 and *p* = 0.0576, respectively). (**E**) In cohort I, CTL activity based on Gr-B release showed that glucomannan at the 2 mg dose as an adjuvant showed a significant increase compared with cell lysate + GM4 mg, lysate vaccine, and PBS groups (*p* = 0.0145, *p* = 0.0001, and *p* = 0.0148, respectively). (**F**) In cohort II, mice treated with glucomannan at 2 mg showed a 58.71, 11.61, and 78.35% increase compared to the 4 mg, PBS, and cyclophosphamide groups (*p* = 0.1095, *p* = 0.6555, and *p* = 0.0511, respectively).

**Figure 4 vaccines-10-01746-f004:**
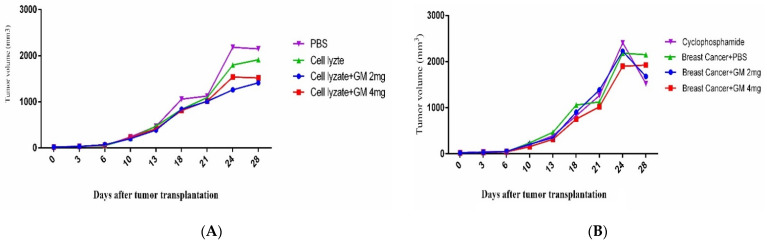
Tumor growth changes in the mouse cohorts: (**A**) In cohort I, the tumor changes on day 28 after tumor transplantation showed that vaccination with cell lysate, cell lysate + GM2 mg, and cell lysate + GM4 mg suppressed tumor growth by 12.27, 41.53, and 52.10% in comparison with PBS-treated mice (*p* = 0.8410, *p* = 0.0418, and *p* = 0.1309, respectively). Immunization with cell lysate + GM2 mg and cell lysate + GM4 mg suppressed tumor growth by 35.43 and 26.01%, respectively, compared to the cell lysate vaccinated group. (**B**) In cohort II, results from tumor volume changed 4 weeks after oral administration with glucomannan at 2 and 4 mg doses, and cyclophosphamide showed 28.00, 11.62, and 41.25% tumor suppression compared with the PBS-treated group, respectively. Furthermore, administration with 2 mg glucomannan showed 14.67% tumor suppression compared to the 4 mg group.

**Figure 5 vaccines-10-01746-f005:**
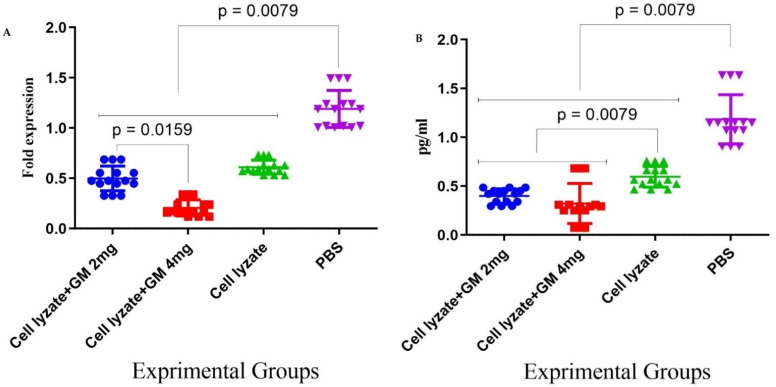
Assessment of adjuvant activity of glucomannan of TGF-β and Foxp3 gene expression in the tumor microenvironment: (**A**) Immunization with the vaccine with glucomannan at the 2 mg dose suppressed Foxp3 gene expression compared to the lysate vaccine and PBS group (*p* = 0.1508 and *p* = 0.0079, respectively). Immunization with cell lysate + GM4 mg showed a significant suppression of Foxp3 gene expression compared with cell lysate + GM2 mg and PBS groups (*p* = 0.0159 and *p* = 0.0079, respectively). In addition, vaccination with cell lysate significantly suppressed Foxp3 gene expression compared with the PBS group (*p* = 0.0079). (**B**) Results for TGF-β gene expression showed that immunization with cell lysate + GM2, 4 mg, and lysate vaccine significantly decreased in comparison with the PBS group (*p* = 0.0079). Immunization with cell lysate + GM2 and 4 mg reduced TGF-β gene expression compared with the lysate vaccine group (*p* = 0.0079).

**Figure 6 vaccines-10-01746-f006:**
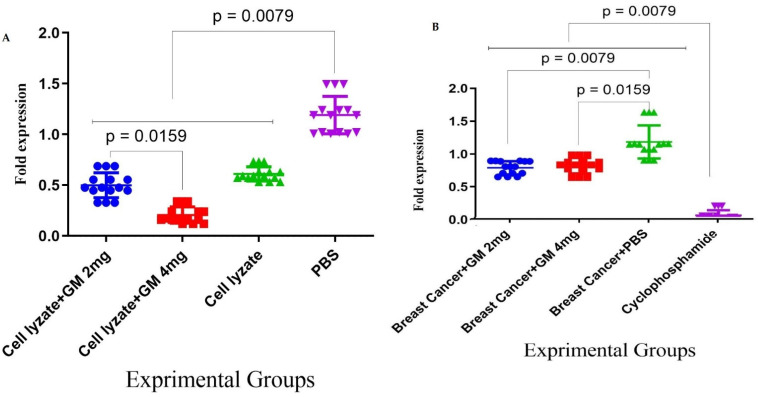
Assessment of the dietary effect of glucomannan on TGF-β and Foxp3 gene expression in the tumor microenvironment: (**A**) Foxp3 gene expression after 4 weeks of oral administration with cyclophosphamide showed a significant decrease in Foxp3 gene expression compared to glucomannan at 2 and 4 mg doses and the PBS group (*p* < 0.05, *p* < 0.05, and *p* = 0.0079, respectively). In addition, administration with glucomannan at 2 and 4 mg showed a significant decrease compared to the PBS group (*p* = 0.0079). (**B**) Results for TGF-β gene expression in the tumor microenvironment showed that, after 4 weeks of oral administration with cyclophosphamide, there was a significant decrease in TGF-β gene expression compared to glucomannan at the doses of 2 and 4 mg and the PBS group (*p* = 0.0079). Administration with 2 and 4 mg doses of glucomannan showed a significant decrease compared to the PBS group (*p* = 0.0079 and *p* = 0.0159, respectively).

**Table 1 vaccines-10-01746-t001:** Experimental groups.

**Cohort-I**			
	**Route of Administration**	**Groups**	**N ^a^**
	subcutaneously	**Group 1**: Naïve mice immunized with 100 µg of tumor lysate vaccine + 2 mg glucomannan.	17
	subcutaneously	**Group 2**: Naïve mice immunized with 100 µg of tumor lysate vaccine + 4 mg glucomannan.	17
	subcutaneously	**Group 3**: Naïve mice immunized with 100µg of tumor lysate vaccine + 100 mg glucomannan.	17
	subcutaneously	**Group 4**: Naïve mice immunized with PBS.	17
**Cohort-II**			
	**Route of Administration**	**Groups**	**N ^a^**
	orally	**Group 1**: Tumor-bearing mice were fed with 2 mg glucomannan.	17
	orally	**Group 2**: Tumor-bearing mice were fed with 4 mg glucomannan.	17
	orally	**Group 3**: Tumor-bearing mice were fed with 200 µg cyclophosphamide daily as the positive control.	17
	orally	**Group 4**: Tumor-bearing mice were fed with PBS buffer daily as the negative control.	17

^a^ N—number.

**Table 2 vaccines-10-01746-t002:** Primer sequences of experimental genes.

Gene	Primer Sequence
**TGF-β**	**Forward**: 5′-CCGCATCTCCTGCTAATGTTG-3 **Revers**: 5′-AATAGGCGGCATCCAAAGC-3′
**Foxp3**	**Forward**: 5′-F: CAGCTGCCTACAGTGCCCCTAG-3′ **Revers**: 5′-CATTTGCCAGCAGTGGGTAG-3′
**β-actin**	**Forward**: 5’-TGGAATCCTGTGGCATCCATGAAAC-3′ **Revers**: 5’-TAAAACGCAGCTCAGTAACAGTCCG-3’

## Data Availability

Not applicable.

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
