# Peer review of "Glucomannan as a Dietary Supplement for Treatment of Breast Cancer in a Mouse Model"

_vaccines, 2022, doi:10.3390/vaccines10101746_

Round 1
Reviewer 1 Report (Previous Reviewer 1)
The authors have answered to all comments. Therefore, the paper can be accepted.
Author Response
Hi dear reviewer
Thank you for reviewing our article and helping us to improve our article .
warm regards
Reviewer 2 Report (New Reviewer)
Ahmadi et al., have submitted the study entitled “Glucomannan for treatment of breast cancer as a dietary supplement in mouse breast cancer model”.
The overall study got a good impression.
However, this reviewer has some suggestions to improve more.
Abstract has several redundant sentences, like “commercial ELISA kits”, “using an ELISA kit”, and others.
Also, the abstract looks a bit lengthy, please keep it as precise as possible.
As the tumour lysate method is a crude way of assessing the Glucomannan response, therefore please discuss more the lysate in the results and discussion sections, like what is the major antigenic determinant in the lysates, what the reported literature says about the specific tumour antigens with Glucomannan, so and so.
I wonder why the authors have not included the most important group, the prophylactic group. E.g., feeding/supplementing Glucomannan to mice for a long period of time and then evaluating the susceptibility of tumour development.
All cytokines data should be represented as one figure with subsections in it.
Figure 4b: “P=0.0379” font size should be maintained uniformly.
Figure 8: Why have the authors not represented the stat here; are they not significant, including cyclophosphamide? Error bars are missing for these figures.
A few of the figures x-axis legends are bold, and a few are not; please be uniform throughout the manuscript.
Author Response
Reviewer 2
Ahmadi et al. have submitted the study entitled “Glucomannan for treatment of breast cancer as a dietary supplement in mouse breast cancer model”.
The overall study got a good impression.
However, this reviewer has some suggestions to improve more.
1- Abstract has several redundant sentences, like “commercial ELISA kits”, “using an ELISA kit”, and others. Also, the abstract looks a bit lengthy, please keep it as precise as possible.
Thank you for this critical comment. The abstract was modified to address the issue better and be shorter. Also, we sent the article for native English review by MDPI editing services to correct the language problems and be better understood and highlighted the correction in yellow.
2- As the tumour lysate method is a crude way of assessing the Glucomannan response, therefore please discuss more the lysate in the results and discussion sections, like what is the major antigenic determinant in the lysates, what the reported literature says about the specific tumour antigens with Glucomannan, so and so.
We should thank you for this vital advice; in discussed, we add information to address the different aspects of the lysates vaccine and its importance in various investigations.
3- I wonder why the authors have not included the most important group, the prophylactic group. E.g., feeding/supplementing Glucomannan to mice for a long period of time and then evaluating the susceptibility of tumour development.
Thank you for this advice; however, regarding it was the results of the Ph.D. project, there were no possibilities for doing IHC and more investigation; however, We try to express these issues in the article and mention this limitation of our investigation.
4- All cytokines’ data should be represented as one figure with subsections in it.
Thank you for this vital comment; we tried categorizing cytocines into TH1, TH2, TH3, and CTL cytokines, so we combined these cytokines to address the main results better.
5- Figure 4b: “P=0.0379” font size should be maintained uniformly.
The font is corrected.
6- Figure 8: Why have the authors not represented the stat here; are they not significant, including cyclophosphamide? Error bars are missing for these figures.
Thank you for mentioning this. In different figures, we showed just significant value, and for mentioned groups, there was no significance we did not show in the figure.
7- A few of the figures x-axis legends are bold, and a few are not; please be uniform throughout the manuscript.
Thank you for a critical review of the figures; we fix these errors in the figures.
Reviewer 3 Report (New Reviewer)
Ahmadi et al.'s paper discusses the potential use of glucomannan as an adjuvant and dietary supplement in mice with grown BrCa tumors. Methodically, paper sounds great. But with respect to the potential application of glucomannan in a clinic, the suppression of less than 30% or 11% sounds insignificant. Also, it sounds as if in the second cohort, table 1's first two lines look similar. Finally, no IHC has been provided to show the activation of FOXP3 or TGFb and downstream targets of those pathways in the mass. As such, my decision will be a minor revision.
Author Response
Ahmadi et al.'s paper discusses the potential use of glucomannan as an adjuvant and dietary supplement in mice with grown BrCa tumors. Methodically, paper sounds great.
1- But with respect to the potential application of glucomannan in a clinic, the suppression of less than 30% or 11% sounds insignificant.
We should express our great apparition for this comment. It is accurate, and regarding that, we should show the most important results in the abstract, and regarding the effect of glucomannan as an adjuvant was more effective, we express those results in the abstract.
2- Also, it sounds as if in the second cohort, table 1's first two lines look similar.
The error was corrected.
3- Finally, no IHC has been provided to show the activation of FOXP3 or TGFb and downstream targets of those pathways in the mass. As such, my decision will be a minor revision.
Thank you for this advice; however, regarding it was the results of the Ph.D. project, there were no possibilities for doing IHC and more investigation; however, We tried to express these issues in the article and mention this limitation of our investigation, highlighting the correction in yellow.
Reviewer 4 Report (New Reviewer)
Glucomannan for treatment of breast cancer; Tumor vaccine adjuvant or immunotherapy supplement? An interesting experience in mouse breast cancer model
The manuscript should be revised for linguistic, grammatical, and style errors.
More details should be provided regarding ethical approval, number, and approved institute.
Histopathology and immunohistochemistry of the target proteins (expressed by the already studied genes) should be evaluated.
Scoring of histopathological pictures and immunohistochemical analysis should be evaluated.
Author Response
Glucomannan for treatment of breast cancer; Tumor vaccine adjuvant or immunotherapy supplement? An interesting experience in mouse breast cancer model
1- The manuscript should be revised for linguistic, grammatical, and style errors.
Thank you for this comment; for this issue, we sent the article for native English review by MDPI editing services to correct the language problems and highlighted the correction in yellow.
2- More details should be provided regarding ethical approval, number, and approved institute.
In the last part of the article, we explain this critical subject. The Ethics Committee approved the animal study protocol of the medical university of Tehran, Iran Ethics number (IR.TUMS.TIPS.REC.1397.12.)
3- Histopathology and immunohistochemistry of the target proteins (expressed by the already studied genes) should be evaluated. Scoring of histopathological pictures and immunohistochemical analysis should be evaluated.
Thank you for this advice; however, regarding it was the results of the Ph.D. project, there were no possibilities for doing IHC and more investigation; however, We tried to express these issues in the article and mention this limitation of our investigation and highlighted the correction in yellow.
Round 2
Reviewer 4 Report (New Reviewer)
As the authors addressed the reviewers' comments, I suggest acceptance of the manuscript.
Author Response
Thank you so much for your comments, as well as the outstanding suggestions and comments made by the reviewer, and make our article more mature and clear.
This manuscript is a resubmission of an earlier submission. The following is a list of the peer review reports and author responses from that submission.
Round 1
Reviewer 1 Report
In this paper, Glucomannan was used in two conditions for cancer immunotherapy, as an adjuvant for breast cancer vaccine model and as a supplement in the immunotherapy of breast tumor bearing mice. The paper can be accepted for the publication after some minor revisions.
1. The novelty of this study must be more explained.
2. Compare your results with others available in the literature.
3. There are some typos. The authors should carefully read the manuscript.
4. Check and unify the citation of references.
Author Response
Dear reviewer,
We thank you for review of our manuscript entitled “Glucomannan for treatment of breast cancer; Tumor vaccine adjuvant or immunotherapy supplement? An interesting experience in mouse breast cancer model’’. Now, we believe that the reviewer's comments make our manuscript mature. The comments were answered carefully (point by point) and were applied in the revised revision of the manuscript. The revised parts were highlighted in yellow.
In this paper, Glucomannan was used in two conditions for cancer immunotherapy, as an adjuvant for the breast cancer vaccine model and as a supplement in the immunotherapy of breast tumor-bearing mice. The paper can be accepted for publication after some minor revisions.
- The novelty of this study must be more explained.
Regarding the novelty of this investigation, our group already showed that the use of tumor lysate vaccine could be a good way to induce an immune response in the breast tumor microenvironment, however, the use of Glucomannan as an adjuvant or supplementary to induce more immune responses to our knowledge is for the first time. The previous studies showed that the uses of Glucomannan not only is an anti-cancer drug by itself but also works as a targeted carrier that is compatible with various bioactive compounds. So that we try to use it in different ways to see can have also an effect as a tumor vaccine adjuvant or not. Also, we mentioned it in the revised version and highlighted it in yellow.
- Compare your results with others available in the literature.
Thank you for these comments. It must be considered that this is the first article that uses Glucomannan as a supplement or tumor vaccine adjuvant in the breast tumor bearing mice. so it could not possible to compare with the literature, however, in the discussion we try to address this issue by comparing our results with other investigations that already use Glucomannan as an adjuvant and also we added available investigations in the revised version and highlighted in yellow.
- There are some typos. The authors should carefully read the manuscript.
Thank you for this comment. The manuscript is completely revised and the errors corrected.
- Check and unify the citation of references.
The reference list was checked and corrected.
Reviewer 2 Report
This work does not allow us to assess what the authors intended. First, the choice of antigens has no basis for the study, because if the tumor induced an immune response, the immune system would have the ability to develop a response against the tumor cells. This has not been verified! On the other hand, a polymer is evaluated as an adjuvant and as a treatment by two different administration routes. In the discussion of the results, the authors compare their results with those published in other studies and, as would be expected, due to the choice of antigen for the vaccine, very different and without great consistency. Thus, this work does not add anything new to what has already been published, so in my opinion it should not be considered for publication.
Author Response
This work does not allow us to assess what the authors intended. First, the choice of antigens has no basis for the study, because if the tumor-induced an immune response, the immune system would have the ability to develop a response against the tumor cells. This has not been verified! On the other hand, a polymer is evaluated as an adjuvant and as a treatment by two different administration routes. In the discussion of the results, the authors compare their results with those published in other studies and, as would be expected, due to the choice of antigen for the vaccine, very different and without great consistency. Thus, this work does not add anything new to what has already been published, so in my opinion it should not be considered for publication.
Dear reviewer
We thank you for review of our manuscript entitled “Glucomannan for treatment of breast cancer; Tumor vaccine adjuvant or immunotherapy supplement? An interesting experience in mouse breast cancer model’’. Regarding the first issue that you mentioned regarding the concern to choose the right antigen for this investigation, it should be mentioned that the use of breast tumor lysate to induce immune responses in the breast tumor microenvironment already has been shown to stimulating the immune response Ashrafi et al. 2017. Besides our previous article, another investigation by Brett P Gross et al .2014 shows that mechanically produced cell fragments hold potential as a preventative vaccine for triple-negative breast cancer. Also, the study by Dombroski et al. 2021 showed that a therapeutic microparticle-based tumor lysate vaccine reduces spontaneous metastases in murine breast cancer. Also, a chaperone protein-enriched tumor cell lysate vaccine generates protective humoral immunity in a mouse breast cancer model. It is proven that tumor lyzate vaccine is not highly immunogen, because of tolerogenic molecules in the tumor lyzate. So, various researchers used this form of vaccine in tumor vaccine studies to evaluate candidate adjuvants in order to increase the vaccine efficacy, as well as our study. In this study, we focused on this fact that in breast tumor bearing mice, in which condition Glucomannan is more bioactive, as vaccine adjuvant and /or as a supplement. And this was the first study aim this.
Reviewer 3 Report
The manuscript entitled "Glucomannan for treatment of breast cancer; Tumor vaccine adjuvant or immunotherapy supplement? An interesting experience in mouse breast cancer model" is well written and organized. But, some issues were raised while reading through the article. I recommend accepting this paper for publication after revising the following suggestions. A few I've suggested below-
1. Some modification is required in the abstract.
2. A graphical abstract is recommended to add highlight the basic concepts of this topic to make a better understanding for the readers
3. The paper contains some grammatical mistakes and syntax errors.
4. The conclusion section should be more specific to this context
Author Response
Dear reviewer,
We thank you for review of our manuscript entitled “Glucomannan for treatment of breast cancer; Tumor vaccine adjuvant or immunotherapy supplement? An interesting experience in mouse breast cancer model’’. Now, we believe that the reviewer's comments make our manuscript mature. The comments were answered carefully (point by point) and were applied in the revised revision of the manuscript. The revised parts were highlighted in yellow.
The manuscript entitled "Glucomannan for treatment of breast cancer; Tumor vaccine adjuvant or immunotherapy supplement? An interesting experience in mouse breast cancer model" is well written and organized. But, some issues were raised while reading through the article. I recommend accepting this paper for publication after revising the following suggestions. A few I've suggested below
- Some modification is required in the abstract.
The Abstract is modified and the correction is highlighted in yellow.
- A graphical abstract is recommended to add highlight the basic concepts of this topic to make a better understanding for the readers
Thank you for this comment. We attached in the revised version.
- The paper contains some grammatical mistakes and syntax errors.
Thank you for your valuable comments. The article was revised and the grammar errors were corrected in the revised version and highlighted in yellow.
- The conclusion section should be more specific to this context.
Thank you for this comment. The conclusion revision and highlighted in yellow.
Round 2
Reviewer 2 Report
Despite the authors' response to my first review of the article, I still consider the lack of importance of the presented study for scientific knowledge. Thus, I maintain my opinion that this article should not be considered for publication.